# Long-term effects of a three-component lifestyle intervention on emotional well-being in women with Polycystic Ovary Syndrome (PCOS): A secondary analysis of a randomized controlled trial

Geranne Jiskoot[1,2,©,¤a]*, Alexandra Dietz de Loos[1,©], Annemerle Beerthuizen[2,©,¤b], Reinier Timman[2,©], Jan Busschbach[2,©], Joop Laven[1,©]

**1** Division Reproductive Endocrinology and Infertility, Department of Obstetrics and Gynecology, Erasmus MC, Rotterdam, The Netherlands, **2** Department of Psychiatry, Section Medical Psychology and Psychotherapy, Erasmus MC, Rotterdam, The Netherlands

© These authors contributed equally to this work.
¤a Current address: Department of Obstetrics and Gynecology, Erasmus MC, Rotterdam, The Netherlands
¤b Current address: Department of Psychiatry, Erasmus MC, Rotterdam, The Netherlands
* L.jiskoot@erasmusmc.nl

## Abstract

Many women with Polycystic Ovary Syndrome (PCOS) report high depression rates. The relationship between PCOS and these high depression rates is unclear. Two-component lifestyle interventions have revealed short-term effects on depression scores in this group of women. In general, 3-component interventions including diet, exercise, and cognitive behavioral therapy (CBT) are more effective in the long-term to improve emotional well-being. This has not yet been studied in women with PCOS. This study examined the effect of 20 CBT lifestyle (LS) sessions combined with a healthy diet and physical therapy with or without 9 months additional feedback through Short Message Service (SMS) via mobile phone, compared to care as usual (CAU, involving advice to lose weight). In this secondary analysis, 155 women with PCOS and a BMI above 25 kg/m$^2$ were eligible. Depression scores decreased significantly in the LS programme compared to CAU (P = 0.045). In both the LS programme without SMS (P = 0.036) and the LS programme with SMS (P = 0.011) depression scores decreased while no change was observed in CAU (P = 0.875). Self-esteem scores improved significantly in the LS programme compared to CAU (P = 0.027). No differences in body image scores were observed in LS participants compared to CAU (P = 0.087), although body image improved significantly in both the LS without SMS (P = 0.001) and with SMS (P = 0.008) study arms. We found no significant mediating role by androgens in the relationship between LS participants and emotional well-being. Only weight-loss mediated the relationship between LS and self-esteem. To conclude, a three-component lifestyle intervention programme with or without additional SMS resulted in significant improvements in depression and self-esteem compared to CAU, in women with PCOS, obesity, and a wish to achieve a pregnancy. Testosterone, androstenedione, DHEA, insulin, HOMA-IR, and cortisol did not mediate this effect. Weight loss mediated the effects

**Data Availability Statement:** All relevant data are within the manuscript and its Supporting Information files.

**Funding:** The author(s) received no specific funding for this work.

**Competing interests:** GJ, ADL, AB, RT and JB have no competing interests. JL has received unrestricted research grants from Ferring, MSD, Merck-Serono, Roche Diagnostics and Euroscreen. He received consultancy fees from the following companies: Euroscreen, Danone-Nutricia, Ferring, Roche Diagnostics, and Titus Healthcare. This does not alter our adherence to PLOS ONE policies on sharing data and materials.

on self-esteem but not on depression and body-image. This suggests that lifestyle treatment independent of weight loss can reduce depression and body-image, but both lifestyle treatment and weight loss can improve self-esteem. Thus, a three-component lifestyle intervention based on CBT could prove successful in improving mood in women with PCOS who are overweight or obese and attempting to become pregnant.

## Introduction

Polycystic Ovary Syndrome (PCOS) is a common endocrine disorder that affects 8–15% of women in their reproductive years [1–3]. The diagnosis of PCOS requires at least two of the following three criteria: (i) oligo-ovulation or anovulation (irregular or no menstrual cycle), (ii) clinical hyperandrogenism (hirsutism) and/or biochemical signs of hyperandrogenism (elevated free androgen index or elevated testosterone levels), (iii) polycystic ovarian morphology (by transvaginal ultrasound), and the exclusion of other etiologies that might cause hyperandrogenism [4]. Most women with PCOS experience one or more of the following physical symptoms in varying degrees: hirsutism (excessive body hair growth), acne, infertility, obesity, insulin resistance and dyslipidemia [5, 6].

Women with PCOS experience more depressive and anxiety complaints, have lower self-esteem, and experience a more negative body image compared to women without PCOS [7–9]. In particular, depression scores are significantly higher [7, 10] and seem to be consistently elevated throughout the lifespan of women with PCOS compared to controls [11]. A recent meta-analysis of depression rates among women with PCOS resulted in a median prevalence of depression of almost 37% compared to 14% in controls [8]. Hence, the recent international guideline on PCOS states that depressive and anxiety symptoms should be screened, assessed and managed with the requirement for awareness of emotional wellbeing [12] Women with PCOS and BMI $\geq$ 30 kg/m$^2$ have significantly higher depression rates compared to women with PCOS and a healthy BMI [10]. A 5% to 10% weight loss improves many PCOS features, including psychological factors [13, 14]. It is unclear how these psychological improvements are generated and whether these psychological improvements are sustained in the long-term. One of the first lifestyle (LS) interventions in women with PCOS was developed by Clark and colleagues. This involved 6 months of seminars covering weight-related topics and resulted in significantly lower depression scores, although no control group was used [15, 16]. Thompson and colleagues developed a 20-week diet and exercise intervention and found significant improvements with respect to depression during the first 10 weeks of the intervention. It is unclear why depression scores did not improve after 10 weeks as participants continued their weight loss and PCOS symptoms improved [17]. A more recent paper demonstrated that a 16-week LS intervention programme resulted in better quality of life [18]. This LS intervention included behavioral modification strategies, although these specific strategies were not described [19]. Others found improvements in depression, health-related quality of life and self-esteem during a high-protein and low-carbohydrate diet, but not in the amount of weight loss [20, 21].

In the general population there is a bidirectional association between obesity and the odds of depression [22]. In women with PCOS the results are inconclusive: some authors concluded that women with PCOS and a higher BMI are more depressed, while others suggest the opposite. Women with PCOS still have higher odds for depressive and anxiety symptoms when matched for BMI [8]. A recent review presented potential mechanisms other than obesity for

the increased depression risk in women with PCOS. Insulin resistance, increased testosterone levels, higher hirsutism scores measured by the modified Ferriman-Gallwey questionnaire, infertility due to oligo-ovulation, increased corticotrophin-releasing hormone, increased cortisol, markers of inflammation, low vitamin D status [23], and elevated Homeostatic Model Assessment for Insulin Resistance (HOMA-IR) levels [24], may contribute to the association between PCOS and depression. Despite the evidence that women with PCOS have increased odds for depression and anxiety, there is no evidence supporting a single etiology for this increased prevalence of depression and anxiety [23]. Thus it remains unclear whether depression is related to one of the above mechanisms and if depression rates could change through weight loss.

The first-line treatment for depression is cognitive behavioral therapy (CBT) and, depending on the setting, can be combined with antidepressant treatment [25]. In the general population, long-term results are mixed; some meta-analysis found CBT to be equally effective compared to other psychological treatments while other meta-analysis found favorable results for CBT [26]. Little research has been undertaken concerning CBT among women with PCOS. A pilot study demonstrated that 8 weeks of 30 minutes' CBT combined with 30 minutes' LS sessions resulted in a significant improvement in quality of life but no improvements in depression scores were observed [27]. A recent randomized control trial (RCT) showed that 8 CBT group sessions of 45 to 60 minutes was effective for psychological fatigue and quality of life [28]. In the new PCOS guideline there is no referral to a specific treatment for depression in women with PCOS, and the advice is to follow regional clinical guidelines [12].

In conclusion, previous studies covered study periods of 24 weeks at most, were not randomized controlled trials, had small sample sizes, and did not use a structured CBT protocol. Hence, we investigated whether a CBT programme for women with PCOS who were overweight and obese achieved weight loss in the long-term in a large sample. The aim of this secondary analysis was to compare the changes in depression scores in a three-component CBT LS intervention (with or without SMS), with these scores in the control group. In addition, the effectiveness of additional SMS on self-esteem and body image was examined. We hypothesized that there is an interaction of androgens (testosterone, androstenedione and dehydroepiandrosterone (DHEA)), insulin, HOMA-IR, and cortisol, on well-being scores in women with PCOS. Hence, we tested whether the relationship between lifestyle treatment and well-being is mediated by changes in androgens (testosterone, androstenedione and DHEA), insulin, HOMA-IR, and cortisol.

## Methods

### Patients

Women were eligible if they were diagnosed with PCOS according to the Rotterdam 2003 consensus criteria, had a BMI above 25 kg/m$^2$, were between 18 and 38 years old, and attempting to become pregnant. Women with inadequate command of the Dutch language, severe mental illness, obesity with another somatic cause, ovarian tumors that lead to an androgen excess, adrenal diseases, had other malformations of their internal genitalia, or those that were pregnant, were not eligible for the study. Participants did not receive any fertility treatment during the study period.

### Study design

This study was approved by the Medical Research Ethics Committee of the Erasmus MC in Rotterdam; reference number MEC 2008–337. The study protocol can be found at http://dx.doi.org/10.17504/protocols.io.bfq9jmz6. The current study on emotional well-being represents

a secondary analysis. At baseline, 183 participants were randomized at a 1:1:1 ratio using a computer-generated random numbers table into three arms: 1) 1-year CBT LS intervention provided by a multidisciplinary team, or 2) 1-year CBT LS intervention provided by a multidisciplinary team extended with a Short Message Service (SMS,) or 3) care as usual (CAU): encouragement to lose weight by publicly available services (i.e. diets, visiting a dietician, going to the gym, or participating in public programmes such as Weight Watchers®). The 1-year multidisciplinary LS intervention aimed at: 1) changing cognitions, 2) changing dietary habits, 3) encouraging and promoting physical activity, and 4) activating social support. It consisted of 20 group sessions of 2.5 hours over one year. During all sessions, CBT techniques were used to create awareness and to restructure dysfunctional thoughts about lifestyle (food and exercise), weight (loss) and self-esteem. More details about the intervention and an overview of the content of each session can be found in the study protocol [29]. Additional to the lifestyle programme, participants in the SMS group sent weekly self-monitored information regarding their diet, physical activity, and emotions by SMS to the psychologist. Subsequently, they received feedback on their messages to provide social support, encourage positive behavior, and empower behavioral strategies.

## Outcome measures

At baseline, and at 3-, 6-, 9- and 12-months, participants attended the outpatient clinic for a standardized screening. This screening included a family and reproductive history, and a physical examination assessing anthropometric and ultra-sonographic features of the syndrome. The primary outcome of the RCT (weight) was also measured. Participants also completed questionnaires on well-being at these time points.

Well-being was measured using three instruments: depression with the Beck Depression Inventory-II (BDI-II), self-esteem with the Rosenberg Self Esteem Scale (RSES), and body image with the Fear of Negative Appearance Evaluation Scale (FNAES).

BDI-II is a validated and widely-used questionnaire in depression trials assessing the severity of depressive symptoms over the previous 2 weeks, according to the DSM-IV criteria. It is a 21-item self-report questionnaire with items rated on a 4-point scale (0–3) and summed to give a total score (range 0–63). A higher score on the BDI-II denotes more severe depression. In non-clinical populations, scores above 20 indicate depression [30]. More specifically: scores of 0–13 indicate minimal depression, 14–19 (mild depression), 20–28 (moderate depression), and 29–63 (severe depression) [31]. The National Institute for Health and Care Excellence (NICE) suggested a difference of ≥3 BDI-II points as a clinically significant effect for normal depression [32]. A recent study estimated a minimal clinically important difference (MCID) for the BDI-II of a 17.5% reduction from baseline [33].

Global self-esteem and self-acceptance was measured by the RSES [34]. This questionnaire consists of 10 questions (5 positive and 5 negative) and has been validated for the Dutch population [35]. Items are rated on a 4-point Likert scale and total scores range from 0 to 30, where a higher score indicates higher levels of self-esteem. There are no official cut-offs, although scores between 15 to 25 are considered as normal self-esteem and scores below 15 as low self-esteem in women with PCOS [36].

The brief version of the FNAES [37] is a short questionnaire consisting of 6 items that measure body image, eating attitude, and depression. The items are answered on a 5-point Likert scale, ranging from 'not at all' to 'extremely', where a higher score indicates more fear of negative evaluation by others (range 6–30). We used a translated version of the FNAES, which has been used before in PCOS [38].

All participants underwent 5 similar standardized measurements during the study period. During these measurements blood samples were collected between 8.00 and 11.00 a.m. after

overnight fasting. Levels of serum testosterone, androstenedione, DHEA, and cortisol were measured with RIA (Siemens) until 2012. After 2012, liquid chromatography-tandem mass spectrometry (LC-MS/MS) was used. Homeostatic assessment of insulin resistance (HOMA-IR) was calculated from fasting insulin and glucose by the following equation: HOMA-IR = (fasting glucose (mg/dl) * fasting insulin (µIU/ml)) / 405 [39].

## Analyses

The power calculation was based on the primary outcome of the lifestyle intervention: weight (kg). The method described by Aberson [25] was applied, with a power of 0.90, a 2-sided alpha of 0.025 (corrected for the interim analysis as described in the study protocol), and 5 repeated measures linearly decreasing. We observed an intercorrelation of around 0.90 between all measurements. With a ratio of 1:1:1, the required sample was 42 in each group. With an expected drop-out proportion of 40% [40], 60 participants in each group were needed for the study.

Descriptive statistics were used to characterize depression, self-esteem, and body image in this sample. Normality of the distributions was checked with Shapiro-Wilks tests. Multilevel regression models were applied for longitudinal analyses of depression (BDI-II), self-esteem (RSES), and body image (FNAES) scores. Mixed modeling can deal efficiently with missing data and unbalanced time-points [41]. This means that, additionally, patients without complete follow-ups could be included in the analyses, without imputation. This method also compensates for selective dropout, on the condition that dropout is related to variables included in the models. The analysis included 2 levels; the patients constituted the upper level and their repeated measures the lower level. The difference from ordinary linear regression is that this analysis takes into consideration that measurements belong to a given participant. The deviance statistic [42] using restricted maximum likelihood [43] was applied to determine the covariance structure, thus taking into account the situation when, e.g., the deviation at baseline was different from the deviations at follow-ups. The covariance structure was determined with deviance tests, using restricted maximum likelihood. To this end, the unstructured component, the variance component and the intercept- only covariance structures were compared amongst each other. In the case of a non-normal distribution a bootstrap procedure with 10,000 samples was performed to obtain reliable standard errors and p-values. Study group, linear and logarithmic time and interactions were included as independent variables. Cohen's d effect sizes were calculated by performing additional multilevel models on normalized outcome measures, using Blom transformations [44]. Blom transformations have the characteristic that they are standardized, thus the outcomes are Cohen's d effect sizes. Cohen valued d = 0.2 as a 'small' effect size, 0.5 as a 'medium' effect size and 0.8 as a 'large' effect size [45]. To test if androgens, insulin, HOMA-IR, and cortisol mediated the effect of LS intervention with or without SMS on emotional well-being, we used multilevel longitudinal mediation or indirect effect analyses. Paths α, β, τ and τ′ were estimated employing multilevel regression analyses. Firstly, we determined whether paths β were significant. When path β was not significant, mediation was improbable. We adjusted the Sobel-Goodman test for the indirect effect of the independent variable on the dependent variable as reported by MacKinnon and Dwyer [46], following the recommendations by Krull and MacKinnon [47] for multilevel mediation analyses. The significance of the mediated effect is given by [48]:

$$Z_{mediation} = \frac{\alpha\beta}{\sqrt{\beta^2 SE_\alpha^2 + \alpha^2 SE_\beta^2 + SE_\alpha^2 SE_\beta^2}}$$

All analyses were performed utilizing IBM Corp (Released 2017. IBM SPSS Statistics for Windows, Version 25.0. Armonk, NY: IBM Corp).

## Results

Between August 2nd 2010 and March 11th 2016, 561 eligible women were asked to participate and 209 provided written informed consent, of whom 26 were included in a pilot study. The total sample for this secondary analysis consisted of 140 women who completed the depression questionnaire, 155 who completed the body image questionnaire, and 141 who completed the self-esteem questionnaire, all at baseline (Fig 1 and Table 1). According to the Shapiro-Wilks test none of the baseline outcome variables were normally distributed. For all mixed models linear time was not significant, thus superfluous. The logarithm of time was included in all the models. The multilevel models are presented in S1 Table.

### Depression

For depression a variance component covariance structure was found to be optimal (S2 Table). Depression scores decreased significantly in the LS intervention compared to CAU (Cohen's d = -0.34; p = 0.045). We observed no difference between LS with SMS and LS without SMS (Cohen's d = -0.02; p = 0.939), Table 2 and Fig 2. Over the study period, depression scores decreased in LS without SMS by 3.7 points (Cohen's d = -0.35; p = 0.036) and in the LS with SMS by 3.8 points (d = -0.37; p = 0.011). The decrease in LS with and without SMS is considered clinically significant, and both LS groups reached the MCID threshold of a more than 17.5% reduction from baseline. Within the CAU group no change in depression scores was observed (d = -0.02; p = 0.875, Table 3).

### Self-esteem

It was also the case that for self-esteem a variance component covariance structure was found to be optimal. Self-esteem scores improved significantly in the LS intervention compared to CAU (Cohen's d = 0.48; p = 0.027, Table 3 and Fig 3). We observed no beneficial effect for additional SMS during lifestyle treatment (Cohen's d = -0.07; p = 0.759). Self-esteem scores improved in the LS intervention without SMS by 2.6 points (Cohen's d = -0.44; p<0.001), and in the LS with SMS by 2.2 points (d = -0.36; p = 0.002). Self-esteem scores remained virtually stable within the CAU group (d = -0.02; p = 0.688), Fig 3.

### Body image

For body image self-esteem an intercept only covariance structure was found to be optimal. We observed no difference for LS intervention compared to CAU (Cohen's d = -0.37; p = 0.087), see Table 3. Although body image scores did improve significantly within the LS intervention without SMS (d = -0.50; p = 0.001) and in LS with SMS (d = -0.47; p = 0.008). The improvement within the CAU group of d = -0.12 was not statistically significant (p = 0.447), Table 2 and Fig 4.

### Mediation

We tested 21 different paths β: the relationship between LS intervention and the 3 outcome measures (depression, self-esteem, and body image) with 7 potential mediators (testosterone, androstenedione, DHEA, insulin, HOMA-IR, cortisol, and weight loss), Fig 5. Only 4 paths β out of 21 turned out to be statistically significant with self-esteem; weight loss, androstenedione, testosterone, and DHEA. No significant paths were observed for insulin, HOMA-IR, and

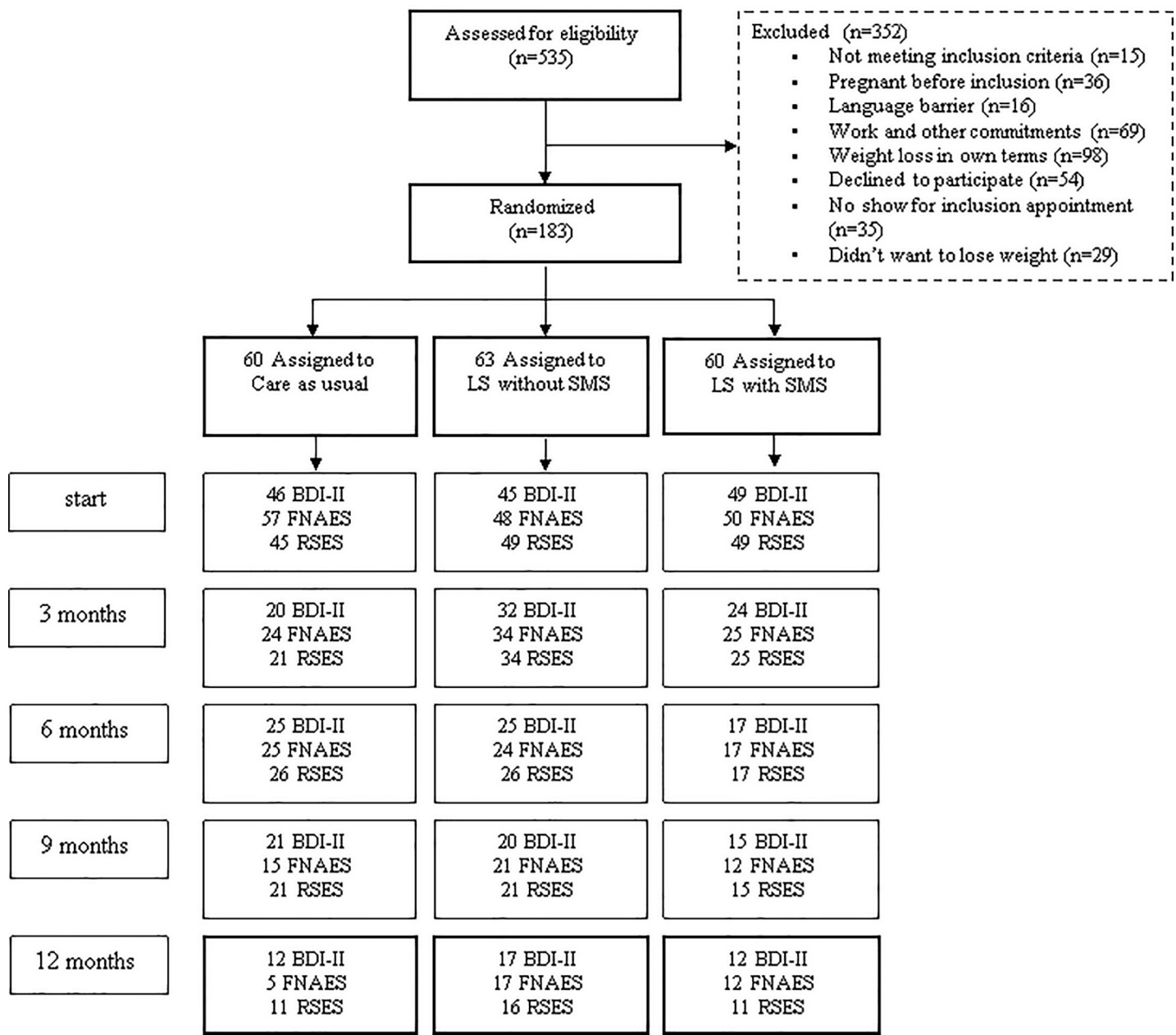

**Fig 1. Consort flowchart.** BDI-II = depression, RSES = self-esteem, FNAES = body image.

cortisol. Consequently, there was no mediation by insulin, HOMA-IR and cortisol. Notably, none of the paths with depression turned out to be statistically significant. When the potential mediators with statistical significant paths β were used, mediation was not found in either the relationship between LS (p = 0.613) and self-esteem with androstenedione, between LS with self-esteem and testosterone (p = 0.834), nor between LS with self-esteem and DHEA (p = 0.737). We also tested if weight loss mediated the effects on depression, self-esteem, and body image. First, we examined weight loss in all groups. In CAU, participants lost 2.32 kg, 4.65 kg in LS without SMS and 7.87 kg in LS with SMS (within all groups P<0.001). Second, we examined mediation in all three well-being outcomes. We found a nearly significant

**Table 1. Baseline characteristics by trial group.** IQR = Interquartile range, BDI-II = depression, RSES = self-esteem, FNAES = body image. Spontaneous pregnancies observed during the study period.

| | Control (CAU) | Lifestyle without SMS | Lifestyle with SMS |
|---|---|---|---|
| | Median [IQR] | Median [IQR] | Median [IQR] |
| BDI-II scores | 11.0 [5.0–18.0] | 13.5 [5.9–24.0] | 12.0 [5.5–20.9] |
| RSES scores | 23.0 [17.5–26.0] | 20.0 [14.0–23.0] | 20.0 [16.0–24.7] |
| FNAES scores | 15.0 [8.0–21.5] | 20.5 [13.3–23.8] | 19.0 [12.0–23.3] |
| Age (year) | 28.0 [26.0–32.0] | 30.0 [27.0–33.0] | 28.0 [26.0–32.0] |
| Attempting to conceive (months) | 27.5 [15.0–59.0] | 27.0 [16.0–63.5] | 24.5 [11.8–36.3] |
| Weight (kg) | 84.0 [79.0–97.3] | 89.0 [80.0–103.5] | 94.5 [85.3–105.8] |
| Height (cm) | 165 [160–170] | 164 [160–169] | 167 [161–170] |
| BMI (kg/m$^2$) | 30.6 [29.3–34.3] | 33.5 [30.4–36.0] | 33.5 [30.9–37.1] |
| Waist (cm) | 96 [89–109] | 100 [93–107] | 102 [94–110] |
| Hip (cm) | 114 [107–122] | 116 [109–124] | 120 [113–129] |
| Waist-Hip ratio | 0.84 [0.80–0.90] | 0.87 [0.81–0.90] | 0.84 [0.80–0.90] |
| Modified Ferriman–Gallwey score | 3 [1–6] | 4 [2–9] | 3 [1–9] |
| Testosterone | 1.48 [1.10–2.00] | 1.55 [1.20–2.20] | 1.49 [0.99–2.00] |
| Androstenedione | 8.8 [5.7–13.8] | 7.7 [5.3–11.0] | 8.5 [5.0–13.4] |
| Dehydro-epiandrosterone (DHEA) | 24.9 [19.1–44.2] | 21.2 [14.3–27.9] | 21.6 [15.2–34.9] |
| Insulin | 88.5 [62.0–122.5] | 102.5 [54.0–147.5] | 87.0 [50.5–122.0] |
| HOMA-IR | 1.10 [0.77–1.57] | 1.26 [0.68–2.01] | 1.10 [0.67–1.65] |
| Cortisol | 309 [248–366] | 262 [220–334] | 323 [237–385] |
| | N (%) | N (%) | N (%) |
| Menstrual cycle | | | |
| *Oligomenorrhea* | 40 (88.9) | 36 (80.0) | 33 (70.2) |
| *Amenorrhea* | 4 (8.9) | 7 (15.6) | 12 (25.5) |
| *Regular* | 1 (2.2) | 2 (4.4) | 2 (4.3) |
| Spontaneous pregnancies | 10 (16.7) | 16 (26.7) | 14 (23.3) |
| Hirsutism | 11 (23.9) | 16 (35.6) | 14 (28.6) |
| Caucasian | 14 (30.4) | 17 (37.8) | 24 (49) |
| Education | | | |
| *Low* | 5 (20.0) | 1 (3.4) | 2 (6.1) |
| *Intermediate* | 15 (60.0) | 16 (55.2) | 20 (60.6) |
| *High* | 5 (20.0) | 12 (41.4) | 11 (33.3) |
| History of depression | 0 (0.0) | 0 (0.0) | 4 (8.2) |
| BDI-II > 13 | 19 (41.3) | 22 (50.0) | 21 (42.9) |
| BDI-II > 20 | 9 (19.6) | 16 (36.4) | 13 (27.1) |

(p = 0.08) relationship between weight loss and self-esteem. In other words, weight loss had a nearly significant effect on the treatment-related changes in self-esteem. Weight loss appeared to be a strong mechanism by which the intervention improved self-esteem. Weight loss had no effect on the relationship between lifestyle treatment and improvements in depression or body-image.

## Discussion

This study is the largest RCT investigating weight loss during a three-component CBT lifestyle intervention, and also the first to investigate long-term effects. All previous studies were LS interventions that lasted between 10 and 24 weeks [15–18, 49] and did not examine well-being in the long-term. We thus performed a secondary analysis of the well-being data that was

**Table 2. Difference in depression, self-esteem and body image changes between study groups at 12 months.** The difference in depression, self-esteem and body image scores over time compared between lifestyle and care as usual and compared between lifestyle with SMS and without SMS. Cohen's D 0.20 is small effect, 0.50 is medium effect and 0.80 a large effect, BDI-II = depression, RSES = self-esteem, FNAES = body image.

| | Lifestyle vs Care as Usual | Lifestyle with SMS vs. Lifestyle without SMS |
|---|---|---|
| | Estimate | Estimate |
| **BDI-II difference** | -3.44 | -1.19 |
| Cohen's d | -0.41 | 0.04 |
| P value | **0.045** | 0.628 |
| **RSES difference** | 2.65 | -0.24 |
| Cohen's d | 0.46 | -0.04 |
| P value | **0.003** | 0.823 |
| **FNAES difference** | -2.60 | 0.63 |
| Cohen's d | -0.29 | 0.13 |
| P value | 0.094 | 0.756 |

collected in the RCT. We observed positive effects of LS treatment on depression scores during the entire 12 months intervention period. Others only observed short term effects that lasted for 10 weeks [17]. As discussed in the Introduction, some researchers have suggested that women with PCOS appear to have a unique risk for depression [8] that is persistent over time [11, 50], which could either be related to the condition itself, or to: weight, androgens, insulin, cortisol [22]. Hence we tested the potential mediation of androgens, insulin, HOMA-IR, and cortisol in the relationship between LS treatment and emotional well-being. Surprisingly, we found neither mediation by androgens nor by insulin, HOMA-IR, or cortisol. A nearly significant relationship was found between LS treatment and self-esteem mediated by weight loss, suggesting that the effects on self-esteem were caused by changes in weight loss. Our results suggest that the three-component intervention was the determining factor with respect to the improvements in depression and body-image, and that improvements in self-esteem were mediated by weight loss.

Compared to other LS interventions performed in women with PCOS [15, 16, 18], our intervention was the only one that was CBT-based. Previous interventions involved seminars covering weight-related topics [16, 49] or behavioral modification strategies [18]. Comparing our intervention to others is difficult because there are large differences in treatment protocols or information is lacking on which behavioral strategies are used. To optimize future research and promote treatment adherence, we used a standardized CBT protocol for all 20 sessions. During every therapy session a given topic was discussed, and the specific CBT techniques for that session were described in the study protocol [29].

In addition to the significant decline in depression scores, we also observed a clinically significant decline of ≥3 points [32] and a minimal clinically important difference (MCID) of more than the threshold of 17.5% [33] in LS interventions with and without SMS. The MCID is the optimal threshold above which individuals report feeling 'better'. In other words, the three-component LS intervention improved depression rates while no changes in depression rates during CAU were observed. Little is known about the possible mechanism through which LS interventions achieve their effects or which components contributed the most [51]. Due to the design of our study we do not know if 1 or 2 of the 3 components (diet, exercise, CBT), or the 3 components as a whole, affected emotional well-being.

Participants in our study had lower mean self-esteem and lower body image scores compared to a previous study in women with PCOS [38]. This difference could be explained by

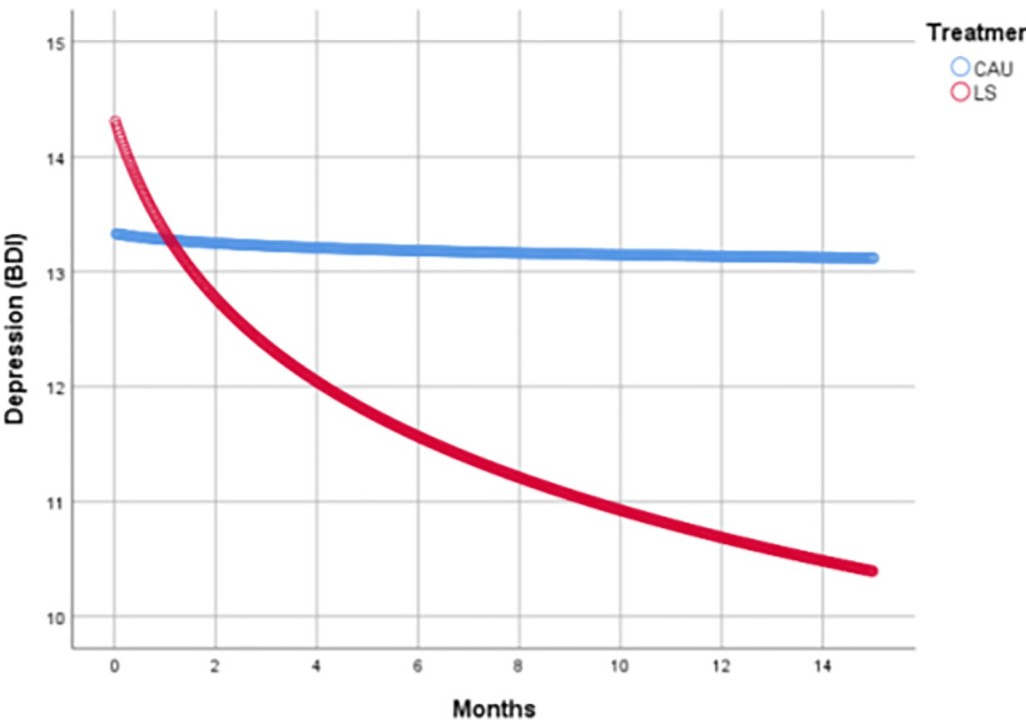

**Fig 2. Depression scores over time.**

BMI because our study population was more obese. As shown in the mediation analysis, weight loss mediated treatment effects on self-esteem, while this was not the case for the changes in depression and body-image during LS treatment. This result is in line with a meta-analysis of well-being outcomes in weight loss treatments. Only treatments that produced actual weight loss showed increased self-esteem, whereas improvements in depression were independent of weight loss. This indicates that self-esteem and depression are different constructs [52]. The improvements in body-image could be caused by the combination of CBT and group treatment, where group cohesion and social support might have played an important role. Many participants mentioned that the LS programme helped them to realize that they 'were not alone', emphasizing that PCOS and obesity made them feel lonely and insecure. It is known that group cohesion and social support can be strong in small groups [53] and

**Table 3. Estimated depression, self-esteem and body image scores over time.** Cohen's D: 0.20 = small effect, 0.50 = medium effect and 0.80 = a large effect.

| | Group | Baseline | 12 months | Change baseline—12 months | | | |
|---|---|---|---|---|---|---|---|
| | | Estimate | Estimate | Estimate | Percent | Cohen's d | P value |
| Depression (BDI-II) | Care as usual (CAU) | 13.3 | 13.1 | -0.2 | -1.5% | -0.02 | 0.875 |
| | Lifestyle without SMS | 15.5 | 11.9 | -3.7 | -23.6% | -0.35 | **0.036** |
| | Lifestyle with SMS | 13.2 | 9.4 | -3.8 | -29.0% | -0.37 | **0.011** |
| Self-esteem (RSES) | Care as usual (CAU) | 21.2 | 20.8 | -0.4 | -1.8% | 0.02 | 0.688 |
| | Lifestyle without SMS | 18.8 | 21.5 | +2.6 | +14.0% | 0.44 | **<0.001** |
| | Lifestyle with SMS | 19.5 | 21.7 | +2.2 | +11.2% | 0.36 | **0.002** |
| Body image (FNAE) | Care as usual (CAU) | 15.5 | 14.6 | -0.9 | -5.5% | -0.12 | 0.447 |
| | Lifestyle without SMS | 18.9 | 15.4 | -3.5 | -18.5% | -0.50 | **0.001** |
| | Lifestyle with SMS | 18.1 | 14.8 | -3.3 | -18.1% | -0.47 | **0.008** |

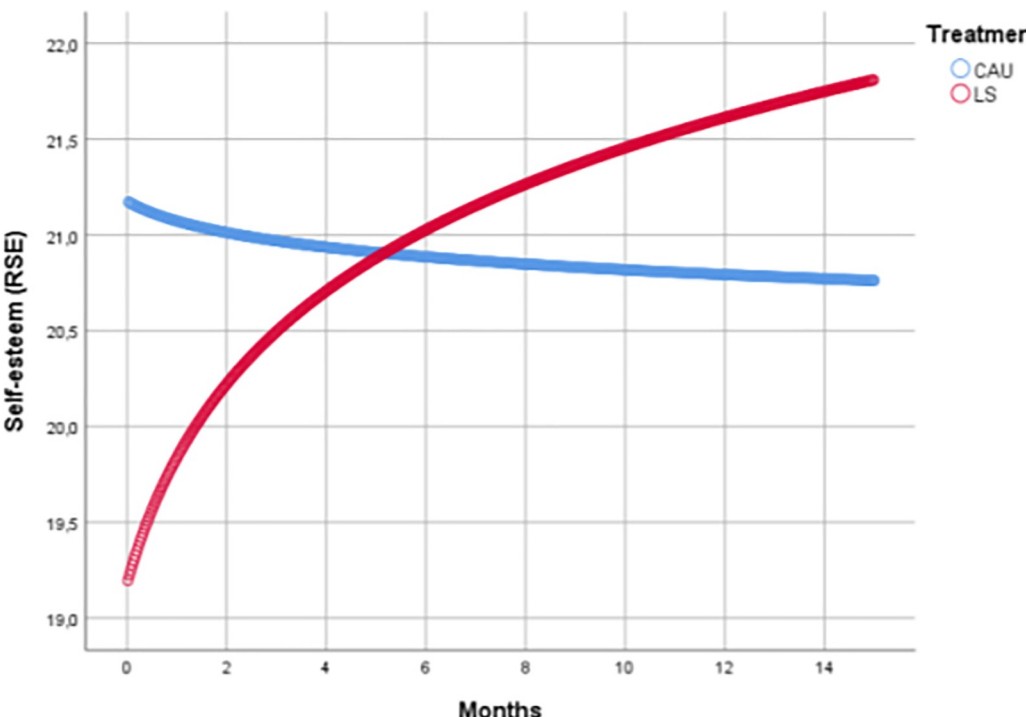

**Fig 3. Self-esteem scores over time.**

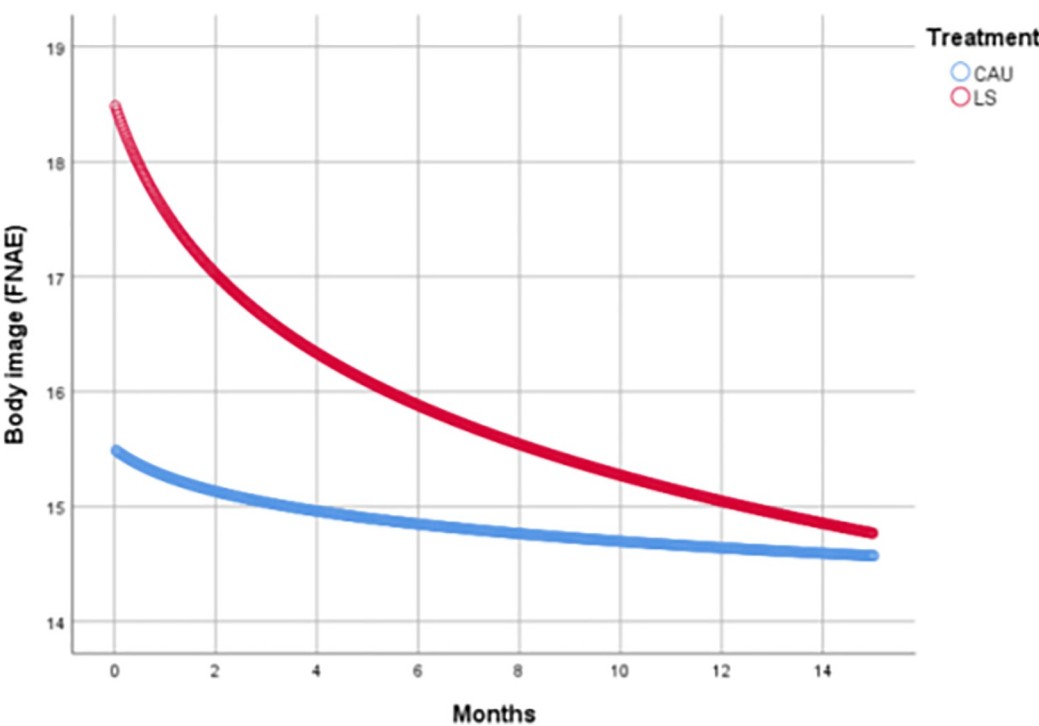

**Fig 4. Body image scores over time.**

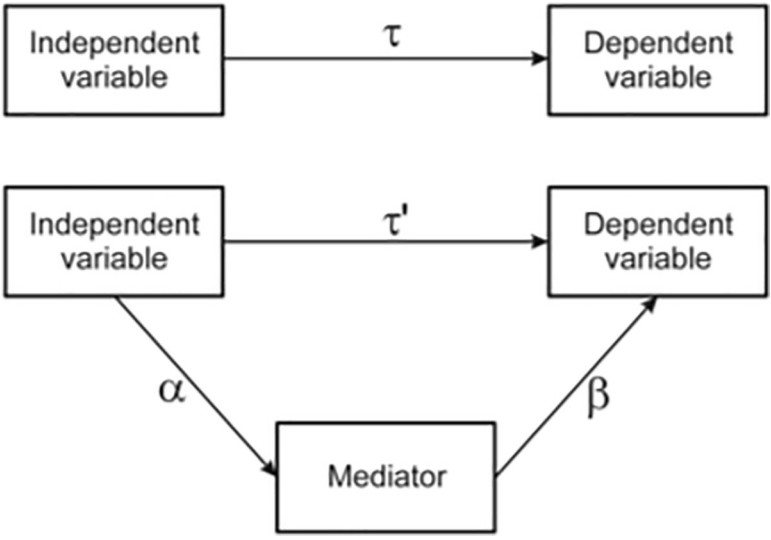

**Fig 5. Mediation effects.**

especially where group members have similar backgrounds [54]. Other researchers have found that, especially for women with PCOS, group support is important for behavior change and reducing social isolation [55, 56]. The combination of small group treatment and one-year treatment seems beneficial for this group of women with PCOS beyond weight loss.

A strength of the current study is that we started with a population that was not severely depressed, whereas other researchers only either included [56] or excluded [57–59] severely depressed patients. [57–59]. Many LS programmes have excluded participants with symptoms of depression based on the idea that they may lose less weight than non-depressed participants [57–59]. Hence, our population might be a reliable reflection of the clinical situation where a substantial number of women with PCOS report moderate depressive symptoms [8]. It has also been suggested that depressed participants should be identified before entering an LS intervention and offered treatment for depression before entering an LS intervention [60]. Based on our findings, we consider that all women with PCOS, depressed or not depressed, can benefit from a three-component LS intervention. Moreover, in particular, participants with elevated depression scores at baseline should be selected for these interventions, since they can benefit the most from lifestyle treatment.

A limitation of our study lies in the high discontinuation rates we observed in all arms of the study. Compliance and drop-out are the most difficult aspects of any weight-reduction intervention, especially in programmes that last over 42 weeks [61]. In general weight loss programmes, dropout rates of around 40% are observed [40]. We expected to have relatively high discontinuation rates for two reasons: firstly, the intervention is demanding for participants (the intervention takes place on Monday afternoons and involves a one-year commitment) and secondly, because pregnancy, which is the ultimate goal for all participants, is considered as a reason to end study participation. Because high drop-out rates were expected in this intervention, a statistical method was chosen that could include all available data without imputation. Hence participants without a complete follow-up could also be included. This method also compensated for selective dropout, on the condition that dropout is related to variables included in the model [41].

Future research should examine whether the current LS programme could be further improved with more PCOS-related topics and/or specific CBT sessions about depressive

thoughts. We have implemented the 3-component lifestyle intervention as standard care at our outpatient clinic to contribute to this development. Weight loss and depression are the biggest health concerns of women diagnosed with PCOS [62]. Based on their experiences, most women are not satisfied with the emotional support and help they receive [12, 62]. Thus, we believe that a three-component lifestyle programme should be accessible for all women with PCOS who are overweight or obese and trying to become pregnant. Three-component lifestyle interventions can contribute to a healthier weight, a better mood, and can enhance self-esteem and body image in women with PCOS.

## Conclusions

A three-component LS intervention programme with or without additional SMS resulted in significant improvements in depression and self-esteem compared to CAU in women with PCOS, obesity, and a wish to achieve a pregnancy. Testosterone, androstenedione, DHEA, insulin, HOMA-IR, and cortisol did not mediate this effect. Weight loss mediated the effects on self-esteem but not on depression and body-image. This suggests that LS treatment independent of weight loss can reduce depression and body-image, whereas both LS treatment and weight loss can improve self-esteem. Hence, a three-component lifestyle intervention based on CBT can be successful in improving mood in women with PCOS who are overweight or obese and attempting to become pregnant.

## Supporting information

**S1 Table. Mixed models.**
(TIFF)

**S2 Table. Deviance tests for determination of linear and log time and covariance structure.**
(TIFF)

## Acknowledgments

We thank the entire PCOS team of the Erasmus MC.

## Author Contributions

**Conceptualization:** Geranne Jiskoot, Alexandra Dietz de Loos, Annemerle Beerthuizen, Reinier Timman, Jan Busschbach, Joop Laven.

**Data curation:** Geranne Jiskoot.

**Formal analysis:** Geranne Jiskoot, Reinier Timman.

**Investigation:** Geranne Jiskoot.

**Methodology:** Reinier Timman, Joop Laven.

**Supervision:** Annemerle Beerthuizen, Reinier Timman, Jan Busschbach, Joop Laven.

**Visualization:** Geranne Jiskoot, Annemerle Beerthuizen, Reinier Timman.

**Writing – original draft:** Geranne Jiskoot, Annemerle Beerthuizen, Reinier Timman, Jan Busschbach.

**Writing – review & editing:** Geranne Jiskoot, Alexandra Dietz de Loos, Annemerle Beerthuizen, Reinier Timman, Jan Busschbach, Joop Laven.

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
