## [Decision Letter · Decision Letter 0]

31 Mar 2020

PONE-D-20-02106

Long-term effects of a three-component lifestyle intervention on emotional well-being in women with Polycystic Ovary Syndrome (PCOS): a randomized controlled trial

PLOS ONE

Dear Mrs Jiskoot,

Thank you for submitting your manuscript to PLOS ONE. After careful consideration, we feel that it has merit but does not fully meet PLOS ONE’s publication criteria as it currently stands. Therefore, we invite you to submit a revised version of the manuscript that addresses the points raised during the review process.

The reviewers have highlighted several issues that need to be addressed including the concern over the drop out rate and the bias that may have resulted.

We would appreciate receiving your revised manuscript by May 15 2020 11:59PM. To enhance the reproducibility of your results, we recommend that if applicable you deposit your laboratory protocols in protocols.io, where a protocol can be assigned its own identifier (DOI) such that it can be cited independently in the future. For instructions see: http://journals.plos.org/plosone/s/submission-guidelines#loc-laboratory-protocols

We look forward to receiving your revised manuscript.

Kind regards,

Stephen L Atkin, MD

Academic Editor

PLOS ONE

Journal Requirements:

2. Please include your tables as part of your main manuscript and remove the individual files. Please note that supplementary tables (should remain/ be uploaded) as separate "supporting information" files

3. We note you have included a table to which you do not refer in the text of your manuscript. Please ensure that you refer to Table 1 in your text; if accepted, production will need this reference to link the reader to the Table.

"GJ, ADL, AB, RT and JB have nothing to declare. JL has received unrestricted research grants from Ferring, MSD, Merck-Serono, Roche Diagnostics and Euroscreen. He received consultancy fees from the following companies: Euroscreen, Danone-Nutricia, Ferring, Roche Diagnostics, and Titus Healthcare."

Reviewers' comments:

Reviewer's Responses to Questions

**Comments to the Author**

1. Is the manuscript technically sound, and do the data support the conclusions?

Reviewer #1: Partly

Reviewer #2: Partly

2. Has the statistical analysis been performed appropriately and rigorously? 

Reviewer #1: Yes

Reviewer #2: I Don't Know

3. Have the authors made all data underlying the findings in their manuscript fully available?

Reviewer #1: Yes

Reviewer #2: No

4. Is the manuscript presented in an intelligible fashion and written in standard English?

Reviewer #1: Yes

Reviewer #2: Yes

5. Review Comments to the Author

Reviewer #1: Results are reported from a clinical study, involving 155 obese women with polycystic ovary syndrome randomized to one of three study arms. It examined the effect of cognitive behavioral therapy lifestyle sessions combined with a healthy diet and physical therapy with or without 9 months additional feedback through SMS compared to care as usual. In women who received lifestyle sessions, depression scores were significantly lower than in controls.

Minor revisions:

1- Abstract: Define SMS.

2- Provide more precise p-values, rather than p< 0.05, p<0.01.

3- Line 170: Typographical error: Normality.

4- Line 172: Indicate the underlying covariance structure used in the mixed modeling the criteria for selecting it.

5-Cite the statistical software used for the analysis.

6- State and justify the study’s target sample size with a pre-study statistical power calculation. The power calculation should include: sample size, alpha level (indicating one or two-sided), minimal detectable difference and statistical testing method.

7- Normality is an underlying assumption for applying the Cohen's d statistic. Indicate if the distribution of the deltas (change from baseline to follow-up) for each outcome was normal.

8- As a limitation, discuss the effects of the high drop-out rate.

Reviewer #2: The study by Jiskoot et al assess the impact of CBT +/- SMS on emotional wellbeing in women with PCOS who suffer from overweight/obesity.

Notes

The study is interesting and examines an important topic. The authors have gone into a lot of efforts in designing and providing the CBT + lifestyle programme. The duration of the study is also much longer compared to other CBT studies and the authors deserve praise for that.

The study has clearly been hit by difficulty in recruitment and high drop-out rate which is unfortunate.

Comments

Main concerns.

- In the published protocol of the study ( published in Reproductive Health, Jiskoot et al. 2017); weight loss was the primary outcome while mental health was secondary outcome. However, the authors present the data in this paper as if it is a new study with mental health as the primary outcome, and weight loss is hardly mentioned. This is a protocol deviation which needs clarification.

- I feel the study will be much better if presented as a whole. Similar to the published protocol. With weight loss as the primary outcome and emotional health as secondary outcome.

- Only women who had a BMI > 25 kg/me and wanted pregnancy were included + Large number of drop outs. This affects generalisation of findings.

- It is not clear how missing data have been handled in particular around participants who got pregnant during the study.

- Data were reported for only some of the time points (baseline, 12 months). It will be better to report data at 3, 6, 9, and 12 months if available.

- Data reported do not match all the outcomes included in the protocol. This raises suspicion of selective reporting.

Other concerns:

Abstract:

- Line 34. Please change ‘’obese women with PCOS’’ to ‘’women with PCOS who have overweight/obesity’’.

- Line 36. Weight loss is mentioned in the conclusion but there is no mention of weight loss in the results section to justify this statement. It reads‘’the three-component intervention and/or weight loss is the determining factor for the improvements in emotional well-being.’’

Introduction

- Line 48. ‘’PCOS women’’. Please change to ‘’women with PCOS’’. The same applies throughout the manuscript.

- Line 50. ‘’abnormal amount of cholesterol derived lipids in their blood (dyslipidemia) characterized by high serum levels of triglycerides and low levels of HDL Cholesterol’’. Please rephrase.

- Line 60. ‘’non-obese women with PCOS’’. Please change to ‘’women with PCOS without obesity’’. The same applies throughout the manuscript.

- Line 100. ‘’overweight and obese women’’. Please change to ‘’women with overweight and obesity’’. Same applies throughout the manuscript.

- Line 101. ‘’The aim of the current study was to compare the changes in depression scores in a three-component CBT lifestyle intervention (with or without SMS), with these scores in the control group’’. This does not fit with the study protocol published in Reproductive Health, Jiskoot et al. 2017 where depression was a secondary outcome and weight loss was primary outcome.

Methods:

- Line 112, 113. Inclusion criteria. ‘’Women diagnosed with Polycystic Ovary Syndrome (PCOS) according to the Rotterdam 2003 consensus criteria and a BMI above 25 kg/m2 and a wish to become pregnant’’. Why only women who wanted to get pregnant were included? Why only women with BMI > 25 kg/m2 were included. Would that have implications on the generalisability of the study findings?

- Line113, 114. Exclusion criteria: ‘’pregnancy at baseline or follow-up’’. What about those with underlying mental health problem or on antidepressants?

- The inclusion/exclusion criteria paragraph is very small. Are there any other criteria not mentioned in paper?

- Study design. Lines 117, 118. ‘’The current study on emotional well-being represents a secondary analysis’’. Needs a bit more clarification. Secondary analysis of what? If this paper is a secondary analysis of another study, this should be made clear from the introduction rather than trying to present the data as a new study?

- Can you please clarify what were the study start and end recruitment dates?

- Other measures of quality of life were mentioned in the protocol but not reported, e.g PCOSQ, SF-36? Why not? Could these be included.

Analysis

- Statistics is better reviewed by a statistician.

- Can you please explain what is meant by intention to treat analysis in this study?

- The three groups were not equal in terms of weight or BMI? Has this been adjusted for?

- What happened to weight during the study?

- Has weight loss been considered as a mediator? If not why? can you consider it?

- Was there any difference in depression between LS+CBT vs. usual care group at baseline?

- Can you please clarify the percentage of patients who completed the programme for each questionnaire (BDI/FNAE/RSE). For example, only 46 participants out of 60 completed the BDI questionnaire at baseline in the usual care group. How many of these 46 participants completed the study?

- How many attended each visit at 3, 6, 9 and 12 months?

- In figure 1. 60 patients were assigned to each treatment group. Only 12 – 17 participants completed the study in each group (70 – 80% drop out). How do you explain the large drop out? How did you deal with missing data on follow up?

- In figure 1. More women in the LS+CBT groups got pregnant than the usual care group. How did you manage this data?

- Did any study participant receive any treatment to improve fertility?

Discussion

- Line 237. ‘’Our results suggest that the effect of the three-component intervention and/or weight loss is the determining factor for the improvement in emotional well-being scores’’. There is no mention on weight loss prior to this in the results section?

- Lines 291, 292. ‘’Therefore, we believe that a three-component lifestyle program should be accessible for all women with PCOS.’’ How can you justify this statement if all participants in the study wanted pregnancy and had a BMI > 25 kg/m2

- The limitation paragraph, should include comments on drop out, and restricting inclusion to women with BMI >25 kg/m2 who wanted pregnancy.

Conclusions:

- Line 296. ‘’obese women’’. Please change to ‘’women with obesity’’. Same applies throughout paper.

- Line 298. ‘’ Instead, the three-component intervention and/or weight loss is the determining factor for the improvements in emotional well-being.’’ Again weight loss is mentioned in the conclusions but not explained in results?

6. PLOS authors have the option to publish the peer review history of their article (what does this mean?). If published, this will include your full peer review and any attached files.

Reviewer #1: No

Reviewer #2: Yes: Dr Hassan Kahal

---

## [Author Response · Author response to Decision Letter 0]

23 Apr 2020

April 22nd, 2020

Dear Dr. Atkin, 

Thank you for reviewing our manuscript entitled ‘Successful weight reduction through a long-term cognitive behavioral therapy lifestyle intervention in polycystic ovary syndrome (PCOS): a randomized controlled trial’. 

We are grateful for the detailed and constructive comments received from the reviewers, as they will help us to improve the document. You will find our specific responses in italics below, together with the original questions and comments of the reviewers, and together with the corresponding textual changes made. In the revised manuscript, we have marked these textual changes in yellow. 

With kind regards, 

On behalf of all the authors,

Geranne Jiskoot, MSc

Division of reproductive medicine, Department of Obstetrics and Gynecology, Erasmus MC

Wytemaweg 80, PO Box 2040, 3000 CA Rotterdam, the Netherlands

Phone: +31 (0) 6 44 184 834, E-mail: l.jiskoot@erasmusmc.nl

Reviewer #1: 

Results are reported from a clinical study, involving 155 obese women with polycystic ovary syndrome randomized to one of three study arms. It examined the effect of cognitive behavioral therapy lifestyle sessions combined with a healthy diet and physical therapy with or without 9 months additional feedback through SMS compared to care as usual. In women who received lifestyle sessions, depression scores were significantly lower than in controls.

Minor revisions:

1- Abstract: Define SMS. 

SMS defined as Short Message Service (SMS) via mobile phone and adjusted the sentence in the abstract. 

2- Provide more precise p-values, rather than p< 0.05, p<0.01.

 All the p-values adjusted more precisely in the abstract. 

3- Line 170: Typographical error: Normality.

 Adjusted. 

4- Line 172: Indicate the underlying covariance structure used in the mixed modeling the criteria for selecting it. 

Text added:

The covariance structure was determined by deviance tests, using restricted maximum likelihood. To this end, the unstructured component, the variance component, and the intercept-only covariance structures were compared with each other.

Text added to the results: 

For all mixed models linear time was not significant, thus superfluous. The logarithm of time was included in all models. 

For depression a variance component covariance structure was found optimal (Appendix B).

For self-esteem a variance component covariance structure was also found optimal.

For body image self-esteem an intercept-only covariance structure was found optimal.

Appendix B. Deviance tests for determination of linear and log time and covariance structure.

 -2 loglikelihood df difference (χ²) df p-value

BDI-depression 

linear + log 2711.087 5 

linear only 2711.138 3 0.051 2 0.975

log only 2718.747 3 7.660 2 0.022

unstructured 2707.467 4 

variance components 2711.138 3 3.671 1 0.055

intercept only 2723.752 2 12.614 1 0.000

Self-esteem 

linear + log 2267.505 5 

linear only 2265.149 3 2.356 2 0.308

log only 2272.762 3 5.257 2 0.072

unstructured 2263.733 4 

variance components 2265.149 3 1.416 1 0.234

intercept only 2269.514 2 4.365 1 0.037

Body image 

linear + log 2433.789 5 

linear only 2432.151 3 1.638 2 0.441

log only 2437.781 3 3.992 2 0.136

unstructured 2430.285 4 

variance components 2432.151 3 1.866 1 0.172

intercept only 2434.132 2 1.981 1 0.159

5. Cite the statistical software used for the analysis. 

 Information added with respect to the statistical software in the Analyses section. 

6. State and justify the study’s target sample size with a pre-study statistical power calculation. The power calculation should include: sample size, alpha level (indicating one or two-sided), minimal detectable difference and statistical testing method.

The original sample size calculation was based on a difference between the groups of 0.45 in terms of Cohen’s d in the primary outcome variable (weight), with a power (1-beta) of 0.80 and an alpha level of 0.05 (two-sided). This resulted in 78 patients being enrolled in the lifestyle group with SMS, 78 patients in the lifestyle group without SMS, and 78 patients in the control group, a total of 234. This number was registered at the Dutch Trial Registry. During an interim power analysis we found an effect of Cohen’s d= 0.10 in the control group, whereas the lifestyle intervention group showed an effect of d= 0.52 (a difference of 0.42). Due to this large effect in the intervention group compared to the control group, we modified the original sample size calculation based on the method described by Aberson (25), with a power of 0.90, a two-sided alpha of 0.025 (corrected for the interim analysis as described in the study protocol), and 5 repeated measures linearly decreasing. We observed an intercorrelation of about 0.90 between all measurements. Maintaining a ratio of 1:1:1, the required sample was 42 in each group. With an expected drop-out proportion of 30% (26), 60 participants in each group were needed for the study. 

This information was published in the study protocol paper. We agree with the reviewer that more information should have been added about the sample size calculation in this paper. A shorter version of the text above has been added. 

7. Normality is an underlying assumption for applying the Cohen's d statistic. Indicate if the distribution of the deltas (change from baseline to follow-up) for each outcome was normal. 

We agree that normality is an important assumption for Cohen's d. Additional analyses were applied, using Blom transformed dependent variables to recalculate the Cohen's effect sizes. These were also added to the statistical method section.

8- As a limitation, discuss the effects of the high drop-out rate. 

We agree that the drop-out rate should receive more attention in the discussion section of the paper. The limitations section of the discussion now reads: 

A limitation in our study lies in the high discontinuation rates we observed in all arms of the study. Compliance and drop-out are the most difficult aspects of any weight-reduction intervention, especially in programs that last over 42 weeks (61). In general weight loss programs, dropout rates of around 40% are observed (62). We expected to have relatively high discontinuation rates for two reasons: firstly, the intervention is demanding for participants (the intervention takes place on Monday afternoons and involves a one year commitment) and secondly, because pregnancy, which is the ultimate goal for all participants, is considered as a reason to end study participation. Because high-drop-out rates were expected in this intervention, a statistical method was chosen that could include all available data without imputation. Hence, participants without a complete follow-up could also be included. This method also compensated for selective dropout, on the condition that dropout is related to variables included in the model (40).

Reviewer #2: 

The study by Jiskoot et al assess the impact of CBT +/- SMS on emotional wellbeing in women with PCOS who suffer from overweight/obesity.

Notes

The study is interesting and examines an important topic. The authors have gone into a lot of efforts in designing and providing the CBT + lifestyle program. The duration of the study is also much longer compared to other CBT studies and the authors deserve praise for that.

The study has clearly been hit by difficulty in recruitment and high drop-out rate which is unfortunate.

Comments

Main concerns.

1) In the published protocol of the study (published in Reproductive Health, Jiskoot et al. 2017); weight loss was the primary outcome while mental health was secondary outcome. However, the authors present the data in this paper as if it is a new study with mental health as the primary outcome, and weight loss is hardly mentioned. This is a protocol deviation which needs clarification. 

We agree with the reviewer that the present presentation is ambivalent. Hence we have changed the title into: Long-term effects of a three-component lifestyle intervention on emotional well-being in women with Polycystic Ovary Syndrome (PCOS): a secondary analysis of a randomized controlled trial. To avoid any suggestion of protocol deviation we have stressed the secondary nature of the analysis in: the introduction, the study design, the description of the outcome measures, the results section, and the first part of the discussion. 

2) I feel the study will be much better if presented as a whole. Similar to the published protocol. With weight loss as the primary outcome and emotional health as secondary outcome. 

Indeed, a presentation of the primary and secondary outcomes together would have its merits. However, it proved difficult to present all relevant aspects of both the primary and secondary outcomes in one paper. Combining primary and secondary outcomes in one paper in a transparent way, with an in-depth discussion of all the findings, would result in a manuscript beyond the conventional word limits of papers. We thus split the report of the investigation. A paper concerning the primary outcome (weight loss) is currently being reviewed. We agree that something should be written about weight loss and so we have added some information on this in the results section. 

3) Only women who had a BMI > 25 kg/me and wanted pregnancy were included + Large number of drop-outs. This affects generalization of findings. 

We agree with both reviewers that more information about drop-outs needs to be addressed by the paper; hence the limitations section in the discussion has been expanded. Information has also been added in the conclusion with respect to BMI and the desire to achieve a pregnancy: A three-component lifestyle intervention program with or without additional SMS resulted in significant improvements in depression and self-esteem compared to CAU in women with PCOS, obesity, and attempting to become pregnant.

4) It is not clear how missing data have been handled in particular around participants who got pregnant during the study. 

We have expanded the multi-level analysis text to explain that drop-out or pregnancy is a problem in a complete cases analysis. The multilevel analysis is thus a more correct measure and includes all the additional data if participants have, e.g., measurements at 3 and 6 months and not at 9 and 12 months. This also implies that patients without complete follow-up can be included in the analyses, without imputation. This method also compensates for selective dropout, on the condition that dropout is related to variables included in the models.

5) Data were reported for only some of the time points (baseline, 12 months). It will be better to report data at 3, 6, 9, and 12 months if available. 

Other lifestyle interventions lasted for 3 to 6 months and did not achieve long-term results. Hence we decided to concentrate on long-term results only, those found at 12 months. As the reviewer requested, information is added in figure 1 concerning the time points 3, 6 and 9 months 

6) Data reported do not match all the outcomes included in the protocol. This raises suspicion of selective reporting.

7) Other measures of quality of life were mentioned in the protocol but not reported, e.g PCOSQ, SF-36? Why not? Could these be included.

We decided to focus on well-being in this paper and thus did not include quality of life measures. In addition, it transpired that the PCOSQ questionnaire was used in the study was not a correct translation. Therefore, we have submitted a new study protocol to the Medical Research Ethics Committee to validate this questionnaire. Moreover, earlier research suggested that generic QoL measures such as the SF-36 were not sensitive enough to measure changes in PCOS symptoms: this was originally the reason to include the PCOSQ. Because of the reported insensitivity of the SF-36, it was decided not to publish the SF-36 data in this paper. 

Other concerns:

Abstract:

8) Line 36. Weight loss is mentioned in the conclusion but there is no mention of weight loss in the results section to justify this statement. It reads ‘the three-component intervention and/or weight loss is the determining factor for the improvements in emotional well-being.’

Information about weight-loss is given in the results section. Based on your advice at point 18, we tested weight loss as a mediator and added this analysis to the paper. 

Introduction

9) Line 50. ‘’abnormal amount of cholesterol derived lipids in their blood (dyslipidemia) characterized by high serum levels of triglycerides and low levels of HDL Cholesterol’’. Please rephrase. 

We have changed this sentence into: Most women with PCOS experience one or more of the following physical symptoms in varying degrees: hirsutism (excessive body hair growth), acne, infertility, obesity, insulin resistance, and dyslipidemia

9) Line 101. ‘’The aim of the current study was to compare the changes in depression scores in a three-component CBT lifestyle intervention (with or without SMS), with these scores in the control group’’. This does not fit with the study protocol published in Reproductive Health, Jiskoot et al. 2017 where depression was a secondary outcome and weight loss was primary outcome.

See also points 1 and 2. We have changed the sentence into: The aim of this secondary analysis was to compare the changes in depression scores in a three-component CBT LS intervention (with or without SMS), with these scores in the control group.

Methods:

10) Lines 112, 113. Inclusion criteria. ‘’Women diagnosed with Polycystic Ovary Syndrome (PCOS) according to the Rotterdam 2003 consensus criteria and a BMI above 25 kg/m2 and a wish to become pregnant’’. Why only women who wanted to get pregnant were included? Why only women with BMI > 25 kg/m2 were included. Would that have implications on the generalisability of the study findings?

When this study commenced, it was already known that being overweight and obesity had a negative impact on PCOS characteristics. The study was initiated to improve PCOS characteristics in a high-risk group of women with PCOS. Based on the newest latest PCOS guideline, lifestyle intervention (preferably multi-component including diet, exercise and behavioral strategies) is recommended in all those with PCOS and excess weight, for reductions in weight, central obesity and insulin resistance and not only in patients that desire a pregnancy. We believe that the results of this study are applicable for all women with PCOS. 

11) Lines 113, 114. Exclusion criteria: ‘’pregnancy at baseline or follow-up’’. What about those with underlying mental health problem or on antidepressants?

12) The inclusion/exclusion criteria paragraph is very small. Are there any other criteria not mentioned in paper?

We did not use mental health scores as an exclusion criterion. Required was a reflection of a normal population of women with PCOS with a BMI > 25, thus minimizing the number of exclusion criteria employed to ensure the generalizability of our findings. 

13) Study design. Lines 117, 118. ‘’The current study on emotional well-being represents a secondary analysis’’. Needs a bit more clarification. Secondary analysis of what? If this paper is a secondary analysis of another study, this should be made clear from the introduction rather than trying to present the data as a new study?

 We agree with the reviewer, see answers to points 1 and 2. 

14) Can you please clarify what were the study start and end recruitment dates? 

The start and end dates are described in the first part of the results: Between August 2nd 2010 and March 11th 2016, 561 eligible women were asked to participate and 209 provided written informed consent, of whom 26 were included in a pilot study.

Analysis

15) Can you please explain what is meant by intention to treat analysis in this study?

We have used the definition of intention to treat by McCoy (2017): all participants who are randomized are included in the statistical analysis and analyzed according to the group they were originally assigned, regardless of what treatment (if any) they received.

McCoy, C. E. (2017). Understanding the intention-to-treat principle in randomized controlled trials. Western Journal of Emergency Medicine, 18(6), 1075.

16) The three groups were not equal in terms of weight or BMI? Has this been adjusted for?

Indeed, the lifestyle intervention group started at a higher overall weight. However, mixed modeling takes into account that participants have different baseline values, as defined in the random intercept. In the covariance structure time was also included as a random effect, allowing for participants to have a different course or slope during the study. In lifestyle intervention, especially, the slope (weight loss or weight gain) was important for each individual. Even when the lifestyle group started with a higher baseline weight and finished at about the same weight, it can be concluded that the SMS+ group, in particular, was more successful because the weight loss slope was steeper. 

17) What happened to weight during the study?

 Information added to the results section. 

18) Has weight loss been considered as a mediator? If not why? can you consider it?

We agree that weight loss as a mediator is an interesting question. Thank you for this idea. The analysis has been added to the paper. 

19) Was there any difference in depression between LS+CBT vs. usual care group at baseline? 

The lifestyle intervention group started with slightly higher depression scores but this difference was not statistically significant between LS without SMS and CAU (P=0.103), LS with SMS and CAU (P=0.701), and LS with SMS and LS without SMS (P=0.189). 

20) Can you please clarify the percentage of patients who completed the program for each questionnaire (BDI/FNAE/RSE). For example, only 46 participants out of 60 completed the BDI questionnaire at baseline in the usual care group. How many of these 46 participants completed the study? How many attended each visit at 3, 6, 9 and 12 months?

Figure one has been adjusted. based on your recommendations. 

21) In figure 1. 60 patients were assigned to each treatment group. Only 12 – 17 participants completed the study in each group (70 – 80% drop out). How do you explain the large drop out? How did you deal with missing data on follow up?

We consider the high drop-out numbers to be normal in interventions for weight-loss and especially in programs that are time-consuming (20 Monday afternoon meetings lasting 2.5 hours over a one-year period). For many women this was difficult to combine with work or other commitments. A lifestyle program with an emphasis on changing behavior was different from what many participants expected. Most expected a traditional weight-loss program with a strict caloric deficit. A behavioral program takes more time, and personal reflection is needed in order to change behavior. Additionally, achieving a pregnancy was a reason for drop-out because weight loss is not recommended during pregnancy. We hypothesize that the high number of drop-outs is a combination of these 3 factors.

22) In figure 1. More women in the LS+CBT groups got pregnant than the usual care group. How did you manage this data?

Currently, pregnancy rates are not available for all the participants in the study and especially for participants who dropped-out. The informed consent form we did not include a statement for pregnancy follow-up. Hence a new study protocol has been submitted to the Medical Research Ethics Committee to allow us to perform the follow-up measurements. Pregnancy data is currently being collected in order to prepare a paper concerning the long-term effects of the lifestyle intervention after 5 years of randomization. 

We agree that spontaneous pregnancy data may be confusing because it did not include all pregnancies. Hence it was decided to exclude the spontaneous pregnancy data from Figure 1. 

23) Did any study participant receive any treatment to improve fertility?

No. This information has been added to the methods section, see line 119. 

Discussion

24) Line 237. ‘Our results suggest that the effect of the three-component intervention and/or weight loss is the determining factor for the improvement in emotional well-being scores’. There is no mention on weight loss prior to this in the results section?

The conclusion has been changed to: 

A three-component lifestyle intervention program with or without additional SMS resulted in significant improvements in depression and self-esteem compared to CAU in women with PCOS, obesity and attempting to become pregnant. Testosterone, androstenedione, DHEA, insulin, HOMA-IR and cortisol did not mediate this effect. Weight loss mediated the effects on self-esteem but not on depression and body-image. This suggests that lifestyle treatment independent of weight loss can reduce depression and body-image, but both lifestyle treatment and weight loss can improve self-esteem. Hence, a three-component lifestyle intervention based on CBT can be successful in improving mood in women with PCOS who are overweight or obese and attempting to become pregnant.

25) Lines 291, 292. ‘’Therefore, we believe that a three-component lifestyle program should be accessible for all women with PCOS.’’ How can you justify this statement if all participants in the study wanted pregnancy and had a BMI > 25 kg/m2 and

Line 298. ‘’ Instead, the three-component intervention and/or weight loss is the determining factor for the improvements in emotional well-being.’’ Again weight loss is mentioned in the conclusions but not explained in results?

We agree with the reviewer that this conclusion should be specific for women with PCOS, who are overweight and trying to conceive. This sentence has been changed to: Hence, we believe that a three-component lifestyle program should be accessible for all women with PCOS who are overweight or obese and trying to conceive. 

26) The limitation paragraph, should include comments on drop out, and restricting inclusion to women with BMI >25 kg/m2 who wanted pregnancy.

Limitation section changed. 

Furthermore, we have changed all sentences in lines 34, 48, 60 and 100 into ‘women with PCOS who have overweight/obesity’ or ‘women with PCOS’.

---

## [Decision Letter · Decision Letter 1]

28 Apr 2020

PONE-D-20-02106R1

Long-term effects of a three-component lifestyle intervention on emotional well-being in women with Polycystic Ovary Syndrome (PCOS): a secondary analysis of a randomized controlled trial

PLOS ONE

Dear Mrs Jiskoot,

Thank you for submitting your manuscript to PLOS ONE. After careful consideration, we feel that it has merit but does not fully meet PLOS ONE’s publication criteria as it currently stands. Therefore, we invite you to submit a revised version of the manuscript that addresses the points raised during the review process.

please address the reviewers comments regarding the spontaneous pregnancy 

We would appreciate receiving your revised manuscript by Jun 12 2020 11:59PM. To enhance the reproducibility of your results, we recommend that if applicable you deposit your laboratory protocols in protocols.io, where a protocol can be assigned its own identifier (DOI) such that it can be cited independently in the future. For instructions see: http://journals.plos.org/plosone/s/submission-guidelines#loc-laboratory-protocols

We look forward to receiving your revised manuscript.

Kind regards,

Stephen L Atkin, MD

Academic Editor

PLOS ONE

Reviewers' comments:

Reviewer's Responses to Questions

**Comments to the Author**

1. If the authors have adequately addressed your comments raised in a previous round of review and you feel that this manuscript is now acceptable for publication, you may indicate that here to bypass the “Comments to the Author” section, enter your conflict of interest statement in the “Confidential to Editor” section, and submit your "Accept" recommendation.

Reviewer #1: All comments have been addressed

Reviewer #2: All comments have been addressed

2. Is the manuscript technically sound, and do the data support the conclusions?

Reviewer #1: (No Response)

Reviewer #2: Yes

3. Has the statistical analysis been performed appropriately and rigorously? 

Reviewer #1: (No Response)

Reviewer #2: I Don't Know

4. Have the authors made all data underlying the findings in their manuscript fully available?

Reviewer #1: (No Response)

Reviewer #2: Yes

5. Is the manuscript presented in an intelligible fashion and written in standard English?

Reviewer #1: (No Response)

Reviewer #2: Yes

6. Review Comments to the Author

Reviewer #1: (No Response)

Reviewer #2: Dear Professor Atkin,

The reviewers have engaged well with the review process and I feel the manuscript is now much clearer.

Comments

1. I feel spontaneous pregnancy during the study is an important information and it is worth mentioning it in the manuscript (it was included in the first submission but excluded in revised version).

2. There has been difference in spontaneous pregnancy rates (during study) between the 3 groups (10 in control group, compared to 14 and 16 in the intervention groups). All women participating in the study wanted fertility, was spontaneous pregnancy during the study considered as a mediator/confounder? If not, could it be included?

3. In the baseline data table; the median weight was 84, 89, and 94.5kg in the control, LS, and LS+SMS groups respectively. While the median BMIs were 30.6, 33.5, and 33.5 kg/m2 respectively. I find it strange that the SL+SMS group was 5kg heavier than SL group but with similar BMI. Can this result be double checked?

7. PLOS authors have the option to publish the peer review history of their article (what does this mean?). If published, this will include your full peer review and any attached files.

Reviewer #1: No

Reviewer #2: Yes: Dr Hassan Kahal

---

## [Author Response · Author response to Decision Letter 1]

11 May 2020

May 11th, 2020

Dear Dr. Atkin, 

Thank you for reviewing our manuscript entitled ‘Long-term effects of a three-component lifestyle intervention on emotional well-being in women with Polycystic Ovary Syndrome (PCOS): a secondary analysis of a randomized controlled trial’. 

We are grateful for the detailed and constructive comments received from the reviewers, as they will help us to improve the manuscript. You will find our specific responses to the questions and comments of the reviewers (in italics) and the corresponding textual changes made. In the revised manuscript, we have marked these textual changes in yellow. 

With kind regards, 

On behalf of all authors,

Geranne Jiskoot, MSc

Division of reproductive medicine, Department of Obstetrics and Gynecology, Erasmus MC

Wytemaweg 80, PO Box 2040, 3000 CA Rotterdam, the Netherlands

Phone: +31 (0) 6 44 184 834, E-mail: l.jiskoot@erasmusmc.nl

1. I feel spontaneous pregnancy during the study is an important information and it is worth mentioning it in the manuscript (it was included in the first submission but excluded in revised version).

We included the information about spontaneous pregnancies in Table 1. The difference between LS without SMS compared to CAU was not significant (P=0.816), LS with SMS compared to CAU (P=0.198) and LS with SMS compared to LS without SMS (P=0.130).

2. There has been difference in spontaneous pregnancy rates (during study) between the 3 groups. All women participating in the study wanted fertility, was spontaneous pregnancy during the study considered as a mediator/confounder? If not, could it be included?

We agree with the reviewer that this could be an interesting possibility. Therefore, we have checked if spontaneous pregnancy mediated the effects of the lifestyle treatment on well-being scores. We found no mediation in the relationship between depression with spontaneous pregnancy (P=0.766), nor between self-esteem with spontaneous pregnancy (P=0.570) and body image with spontaneous pregnancy (P=0.890). 

3. In the baseline data table; the median weight was 84, 89, and 94.5kg in the control, LS, and LS+SMS groups respectively. While the median BMIs were 30.6, 33.5, and 33.5 kg/m2 respectively. I find it strange that the SL+SMS group was 5kg heavier than SL group but with similar BMI. Can this result be double checked?

We double checked the data and included the SPSS output for weight, height and BMI below. The median weights as reported in the paper are correct. The LS with SMS is taller and heavier, therefore they have the same BMI as LS without SMS. We have added information about height in Table 1. 

Statistics

group HEIGHT WEIGHT BMI

1 Controle N Valid 60 60 60

 Missing 0 0 0

 Percentiles 25 160.00 79.00 29.336

 50 165.00 84.00 30.662

 75 170.00 97.25 34.375

2 SMS- N Valid 63 63 63

 Missing 0 0 0

 Percentiles 25 160.00 80.00 30.469

 50 164.00 89.00 33.695

 75 169.00 105.00 36.506

3 SMS+ N Valid 60 60 60

 Missing 0 0 0

 Percentiles 25 161.25 85.25 31.227

 50 167.00 94.50 33.577

 75 170.00 105.75 37.071

---

## [Editor Report · Decision Letter 2]

15 May 2020

Long-term effects of a three-component lifestyle intervention on emotional well-being in women with Polycystic Ovary Syndrome (PCOS): a secondary analysis of a randomized controlled trial

PONE-D-20-02106R2

Dear Dr. Jiskoot,

We are pleased to inform you that your manuscript has been judged scientifically suitable for publication and will be formally accepted for publication once it complies with all outstanding technical requirements.

With kind regards,

Stephen L Atkin, MD

Academic Editor

PLOS ONE
---

## [Editor Report · Acceptance letter]

21 May 2020

PONE-D-20-02106R2 

Long-term effects of a three-component lifestyle intervention on emotional well-being in women with Polycystic Ovary Syndrome (PCOS): a secondary analysis of a randomized controlled trial 

Dear Dr. Jiskoot:

I am pleased to inform you that your manuscript has been deemed suitable for publication in PLOS ONE. Congratulations! Your manuscript is now with our production department. 

With kind regards,

on behalf of

Dr. Stephen L Atkin 

Academic Editor

PLOS ONE